# A Three-Wavelength Optical Sensor for Measuring the Multi-Particle-Size Channel Mass Concentration of Thermal Power Plant Emissions

**DOI:** 10.3390/s24051424

**Published:** 2024-02-22

**Authors:** Xiao Xiao, Ming Zhu, Qiuyu Wang, Xiaodong Yuan, Mengxue Lin

**Affiliations:** 1Hubei Key Laboratory of Smart Internet Technology, School of Electronic Information and Communications, Huazhong University of Science and Technology, Wuhan 430074, China; xiaoxiao_dx@hust.edu.cn (X.X.); zhuming@hust.edu.cn (M.Z.); m202272499@hust.edu.cn (Q.W.); m201771997@hust.edu.cn (X.Y.); 2National Engineering Research Center of Fire and Emergency Rescue, Wuhan 430074, China

**Keywords:** three wavelengths, mass concentration, particulate matter, particle size distribution, multi-particle-size channels, thermal power plant

## Abstract

Emissions from thermal power plants have always been the central consideration for environmental protection. Existing optical sensors in thermal power plants usually measure the total mass concentration of the particulate matter (PM) by a single-wavelength laser, bearing intrinsic errors owing to the variation in particle size distribution (PSD). However, the total mass concentration alone cannot characterize all the harmful effects of the air pollution caused by the power plant. Therefore, it is necessary to measure the mass concentration and PSD simultaneously, based on which we can obtain multi-particle-size channel mass concentration. To achieve this, we designed an optical sensor based on the three-wavelength technique and tested its performance in a practical environment. Results showed that the prototype cannot only correctly measure the mass concentration of the emitted PM but also determine the mean diameter and standard deviation of the PSDs. Hence, the mass concentrations of PM_10_, PM_2.5_, and PM_1_ are calculated, and the air pollutants emission by a thermal power plant can be estimated comprehensively.

## 1. Introduction

Environmental pollution is becoming more and more serious with social development. With the increasing concern and demand for environmental protection, the pressure on environmental protection is rising [1]. As an important pollutant in emissions, particulate matter (PM) [2] affects human health in a way related not only to its concentration but also closely to its particle size [3,4]. Studies have shown that as the size of the PM decreases, it becomes exceedingly harmful to the human body owing to the high penetrability of the smaller particles through the alveoli into the blood [5,6]. For example, PM_2.5_ [7] can enter the alveoli and PM_1_ [8] can even directly enter the bloodstream, while PM_10_ is blocked from entering the lungs [9]. In short, PM in air pollutants can cause significant harm to human health through respiratory inhalation, leading to lung cancer or other heart and respiratory diseases. As a main source of air pollution that the Environmental Protection Administration focuses on, thermal power plants have emitted hundreds of thousands of tons of PM annually nationwide [10]. A case study of a thermal power plant in China shows [11] that the mass concentration of PM_2.5_ accounts for about 86% of PM_10_ after wet desulfurization and electrostatic precipitation. It is obvious that the main component of the emissions is PM_2.5_. Therefore, for continuous monitoring of the thermal power plant PM emissions, the measurement of PM_2.5_ must be considered particularly.

However, existing emission monitoring technology just measures the total mass concentration of PM [12,13] and cannot give the mass concentration with PSD. As the mainstream of the existing emission monitoring technology [14,15], photoelectric sensing technology has the advantages of fast response and high sensitivity [16,17,18], but the measurements are easily affected by a change in particle size. Tapered element oscillating microbalance (TEOM) [19] and beta ray technology [20] have high accuracy in measuring the mass concentration of PM, but their sampling time is too long and needs manual maintenance. Considering the high temperature and humidity of the flue smoke emitted by the thermal power plant [21], the common PSD measurement technologies used in laboratory research are difficult to apply in a practical environment. Additionally, the differential mobility analyzer (DMA) method makes it difficult to measure a particle size greater than 1μm due to the limit of voltage [22]. The aerodynamic method cannot measure particles less than 0.5μm because of the response threshold limit [23], and the measurement range of quantity concentration is low. In summary, present PSD measuring methods cost too much time and maintenance accessories or have measurement limits in particle size, and thus cannot be applied in continuous monitoring of PM at the thermal power plant. Due to the lack of accurate particle size and mass concentration monitoring methods, it is currently difficult to accurately and comprehensively evaluate the harmful effects of PM emitted from thermal power plants. Therefore, it is necessary to develop new monitoring technology related to particle size and provide solid support for environmental protection.

In this study, we focused on the technical challenges of continuous emission monitoring of high-temperature and high-humidity flue smoke emitted from thermal power plants. A reliable optical sensor is designed and implemented based on the three-wavelength laser light sources for measuring the multi-size-channel mass concentration of the emissions from the power plant. The contributions of this study are as follows. We proposed a retrieval method based on the three-wavelength technique, which can measure the mass concentration of PM_10_, PM_2.5_, and PM_1_ simultaneously. We designed the structure of an optical sensor, which suits the alignment of the optical paths and avoids interference between each optical path. We applied and tested the sensor in a practical thermal power plant, and the results showed that the sensor can correctly measure the mass concentration in multi-particle-size channels. Thus, we estimated the harmful effect of emitted PM comprehensively.

This paper is organized as follows. Section 2 introduces the proposed three-wavelength method with a scattering intensity model. Based on the method, Section 3 further describes the three-wavelength sensor design in detail. Section 4 shows the practical test results of the prototype sensor in a thermal power plant, followed by the conclusions in Section 5.

## 2. Proposed Three-Wavelength Based Measurement Method

In this section, we described a method that can retrieve the mass concentration of multi-size channels based on the PSD statistical information obtained by the three-wavelength method [24]. First, a known distribution function is assumed according to the literature about PSD measured in thermal power plants. Then, a light scattering model is developed to obtain the parameters of the distribution function to calculate the mass concentration of PM_10_, PM_2.5_, and PM_1_.

### 2.1. Mass Concentration Measurement Based on Particle Size

Existing optical sensors in thermal power plants merely measure the total mass concentration using a single-wavelength laser. However, the mass concentration of the aerosol $C_{m}$ can be defined as the product of volume concentration $C_{V}$ and density ρ. According to the “three-region” law developed by Gebhart [25], when the wavelength of the incident light is close to the particle size, the volume concentration $C_{V}$ is proportional to the scattering light power $P\left(\lambda_{V}\right)$ as follows:(1)$$C_{m}=\rho \cdot C_{V}=\rho \cdot T_{V}\cdot P\left(\lambda_{V}\right)$$
where $T_{V}$ is the conversion vector. It is obvious that when the PSD of the aerosol changes, the conversion vector should be changed accordingly [26]. This indicates that the single-wavelength method for measuring mass concentration is not accurate when the PSD changes. Therefore, a three-wavelength method is introduced to calibrate the effect of PSD variation and obtain a more accurate conversion vector.

### 2.2. Retrieval of the PSD Based on Three Wavelengths

The PSD of emitted PM from the thermal power plant has been investigated according to the literature [11,27]. The number concentration distributions of the PM are usually single peaks, and the peaks are typically within 1 μm. Although the mean particle size and the width of distribution may vary in these distributions, it is reasonable to approximate the distributions by a log-normal distribution function, which is described as follows:(2)$$f\left(x,\mu ,\sigma \right)=\frac{1}{\sqrt{2\pi }\cdot \sigma x}\cdot e^{-\frac{{\left(\log x-\log\mu \right)}^{2}}{2\sigma^{2}}}$$
where x is the diameter of the particles and σ and μ are standard deviation and mean diameter of the PSD. Based on classical Mie theory [28], the scattering light power *P* can be expressed as follows:(3)$$P=C_{N}\int f\left(x,\mu ,\sigma \right)\cdot q\left(x,m,\lambda ,\theta \right)\;dx$$
where $q\left(x,m,\lambda ,\theta \right)$ is the light intensity scattered by a particle from an incident light at a certain observing angle θ, λ is the wavelength of the incident light, and m is the refractive index of the particle. Since the measured aerosols are known to emit flue smoke, the refractive index m is assumed to be a constant. In addition, based on the designed optical structure, the observing angle θ is also a known quantity. Therefore, there are only three unknowns, μ, σ, and λ, in Equation (3).

Based on the previous work, the Sauter Mean Diameter (SMD) of the aerosol, $D_{S}$, can be measured by the dual-wavelength method [29]. The SMD is a statistical parameter used to characterize the size of the particles in the aerosol and can be equivalent to the mean diameter μ and is given by [30]
(4)$$\mu \approx D_{S}=6\cdot \frac{C_{V}}{C_{S}}$$
where $C_{V}$ and $C_{S}$ are the volume concentration and surface area concentration of the PSD, respectively.

As described in the “three-region” law [25,31], when the wavelength of the incident light is in the region that is close to the particle size, $q\left(x,m,\lambda ,\theta \right)$ is proportional to $C_{V}$, while the wavelength is in the region that much less than the particle size, and $q\left(x,m,\lambda ,\theta \right)$ is proportional to $C_{S}$. Hence, $C_{V}$ and $C_{S}$ can be obtained by measuring the scattering light power $P\left(\lambda_{V}\right)$ and $P\left(\lambda_{S}\right)$ with appropriate incident light wavelengths $\lambda_{V}$ and $\lambda_{S}$, respectively [24]. Then, the SMD can be calculated by $P\left(\lambda_{V}\right)$ and $P\left(\lambda_{S}\right)$, and is given by [32]
(5)$$D_{S}=T_{\mathrm{SMD}}\cdot \frac{P\left(\lambda_{V}\right)}{P\left(\lambda_{S}\right)}$$

Based on the dual-wavelength method for measuring the mean diameter, obtaining both μ and σ using three wavelengths is naturally derived by mathematical logic. Assuming that three wavelengths $\lambda_{1}$, $\lambda_{2}$, and $\lambda_{3}$ are selected, where $\lambda_{1}<\lambda_{2}<\lambda_{3}$, Equation (3) can be rewritten as follows:(6)$$\left\{\begin{array}{c} P\left(\lambda_{1}\right)=C_{N}\int f\left(x,\mu ,\sigma \right)\cdot q\left(x,m,\lambda_{1},\theta \right)dx\\ P\left(\lambda_{2}\right)=C_{N}\int f\left(x,\mu ,\sigma \right)\cdot q\left(x,m,\lambda_{2},\theta \right)dx\\ P\left(\lambda_{3}\right)=C_{N}\int f\left(x,\mu ,\sigma \right)\cdot q\left(x,m,\lambda_{3},\theta \right)dx \end{array}\right.$$
where $\lambda_{1}=\lambda_{S}$ and $\lambda_{3}=\lambda_{V}$, $q\left(x,m,\lambda_{1},\theta \right)$ and $q\left(x,m,\lambda_{3},\theta \right)$ correspond to the surface area and volume concentration of the aerosol, which is as follows:(7)$$\left\{\begin{array}{c} P\left(\lambda_{1}\right)=\frac{1}{\pi }T_{III}\cdot C_{N}\int f\left(x,\mu ,\sigma \right)\cdot \pi x^{2}dx=\frac{1}{\pi }T_{III}\cdot C_{S}\\ P\left(\lambda_{3}\right)=\frac{6}{\pi }T_{II}\cdot C_{N}\int f\left(x,\mu ,\sigma \right)\cdot \frac{\pi }{6}x^{3}dx=\frac{6}{\pi }T_{II}\cdot C_{V} \end{array}\right.$$
therefore, the mean diameter μ is obtained as follows:(8)$$\mu =6\cdot \frac{C_{V}}{C_{S}}=\frac{T_{III}\cdot P\left(\lambda_{3}\right)}{T_{II}\cdot P\left(\lambda_{1}\right)}$$
since $\lambda_{2}$ is between $\lambda_{1}$ and $\lambda_{3}$, for the particles larger than $\lambda_{2}$, the scattering light power can be approximated to surface area concentration, while for particles smaller than $\lambda_{2}$, the scattering light power can be approximated to volume concentration. $P\left(\lambda_{2}\right)$ is given as follows:(9)$$P\left(\lambda_{2}\right)=\frac{1}{\pi }{T^{\prime }}_{III}\cdot C_{N}\int_{\lambda_{2}}^{+\infty }f\left(x,\mu ,\sigma \right)\cdot \pi x^{2}dx+\frac{6}{\pi }{T^{\prime }}_{II}\cdot C_{N}\int_{-\infty }^{\lambda_{2}}f\left(x,\mu ,\sigma \right)\cdot \frac{\pi }{6}x^{3}dx$$

By introducing $\Phi \left(\frac{\log x-\log\mu }{\sigma }\right)$ as the cumulative distribution function of $f\left(x,\mu ,\sigma \right)$, $P\left(\lambda_{2}\right)$ can be rewritten as follows:(10)$$P\left(\lambda_{2}\right)=\frac{1}{\pi }{T^{\prime }}_{III}\cdot \left[1-\Phi \left(\frac{\log\lambda_{2}-\log\mu }{\sigma }\right)\right]\cdot C_{S}+\frac{6}{\pi }{T^{\prime }}_{II}\cdot \Phi \left(\frac{\log\lambda_{2}-\log\mu }{\sigma }\right)\cdot C_{V}$$
where ${T^{\prime }}_{II}$ and ${T^{\prime }}_{III}$ are the conversion vectors at $\lambda_{2}$. Then. The ratio of scattering light power $P\left(\lambda_{2}\right)$ to $P\left(\lambda_{1}\right)$ is given as follows:(11)$$\frac{P\left(\lambda_{2}\right)}{P\left(\lambda_{1}\right)}=\frac{{T^{\prime }}_{III}}{T_{III}}\cdot \left[1-\Phi \left(\frac{\log\lambda_{2}-\ln\mu }{\sigma }\right)\right]+\frac{{T^{\prime }}_{II}}{T_{II}}\cdot \Phi \left(\frac{\log\lambda_{2}-\log\mu }{\sigma }\right)\cdot \mu$$

It can be concluded that the information on the mean diameter μ and standard deviation σ can be obtained through three appropriate wavelengths of the incident light. In order to validate the relationship between the scattering light power, μ and σ, a simulation based on Mie theory is performed, and the parameter settings are shown in Table 1.

The range of the parameters μ and σ covers the most scenarios of a thermal power plant, and each pair of them generates a log-normal distribution $f\left(x,\mu_{i},\sigma_{j}\right)$, where $\mu_{i}=100\cdot i$ and $\sigma_{j}=1+0.1\cdot j$, in which i=1,2,⋯,50 and j=1,2,3,4,5. The wavelengths of the incident light are selected as 450 nm, 940 nm, and 1550 nm according to the conditions of the “three-region” law combined with the practical laser sources available.

We defined the scattering light power measured after passing through aerosols with different distributions $f\left(x,\mu_{i},\sigma_{j}\right)$ for each wavelength of incident light as $P_{450}^{ij}$, $P_{940}^{ij}$, and $P_{1550}^{ij}$, respectively. To lower the effect of changes in number concentration $C_{N}$ during the measurement, we further calculate the ratios between each scattering light power: $P_{940/450}^{ij}=P_{940}^{ij}/P_{450}^{ij}$, $P_{1550/940}^{ij}=P_{1550}^{ij}/P_{940}^{ij}$, and $P_{1550/450}^{ij}=P_{1550}^{ij}/P_{450}^{ij}$. Figure 1 shows the simulation result of the scattering light power ratios for each $f\left(x,\mu_{i},\sigma_{j}\right)$. The points in the same color represent distributions in different standard deviations from 1.1 to 1.5, while each one of them is for different mean diameters from 100 nm to 5 μm.

Figure 1 shows that each ratio sets $\left(P_{940/450}^{ij},P_{1550/940}^{ij},P_{1550/450}^{ij}\right)$ represent the coordinates of one point in the 3D space. Therefore, when the scattering light power of three wavelengths is measured, the ratio sets $\left(P_{940/450}^{m},P_{1550/940}^{m},P_{1550/450}^{m}\right)$ will be calculated and determine the position of a point in this 3D space. By calculating the distance between the measured ratio point $\left(P_{940/450}^{m},P_{1550/940}^{m},P_{1550/450}^{m}\right)$ and each $\left(P_{940/450}^{ij},P_{1550/940}^{ij},P_{1550/450}^{ij}\right)$, we can find the closest points from the simulation data and thus retrieve the $\mu_{m}$ and $\sigma_{m}$ of the measured aerosol’s PSD. To further minimize the measurement error, we can select the five closest points to the measured ratio point and calculate an average value of $\mu_{i}$ and $\sigma_{j}$ as the mean diameter and standard deviation of the measured aerosol [24].

## 3. Design of the Three-Wavelength Optical Sensor

We first designed a sensor prototype based on the three-wavelength method. Then, the prototype was tested in a laboratory environment, and the result showed that the prototype could measure the mass concentration, SMD, and PSD correctly [24]. A diagram of this original prototype is shown in Figure 2. In Figure 2a, there are three laser sources inserted at A, B, and C, while A′, B′, and C′ are three quartz rods transmitting the scattering lights corresponding to three laser sources. The laser sources are concentrated in order to minimize the difference of measuring areas in each light path. However, when applied to the practical environment, the prototype met practical issues, and a new design was needed.

### 3.1. Optical Structure

In the first version of the prototype, the three laser light sources were concentrated as close as possible in order to limit the size of the measuring area. When the sensor was applied in the laboratory environment, the aerosols were at normal atmospheric temperature and humidity. However, in thermal power plants, the flue smoke is in a state of high temperature and saturated humidity. In order to ensure that the flue smoke does not condense in the measuring area, it is necessary to heat the aerosols in order to keep the temperature above the dew point. Meanwhile, the shell of the prototype should be extended long enough to ensure that the laser sources and circuit unit are not affected by the high temperature and humidity through the measuring area. However, the laser sources require precise installation and alignment. In the first version of the prototype, the three laser sources are put as close as possible considering the size of the collimating lens. Therefore, when the shell extended and the optical path length increased, any tiny adjustment of the laser source at the laser socket would become a large shift at the measuring area. The complexity of the alignment would become extremely high. In order to solve the issue, a new design for the prototype is needed.

The new optical structure is shown in Figure 3. The three laser sources are divided into three parallel optical paths, and each path has a reflecting part at the end for collecting and reflecting the scattering lights. Therefore, the alignment of each laser source is independent, and the stability and robustness of the optical structure are increased. However, the parallel optical paths have caused another issue: the measuring areas are far apart in each path and the measured aerosols may not be consistent at the same time. In order to reduce the systematic errors caused by parallel optical paths, the distance between each path needs to be minimized as much as possible while avoiding the scattering light interference from other optical paths.

The vertical view of the measuring area and reflecting part without a shell is shown in Figure 4. Parts 1 and 5 are the measuring areas in the top and bottom optical path where the incident light is scattered by measured aerosol. Parts 2, 4, and 6 are apertures on the baffle in each path. Parts 3 and 7 are ring-like reflectors in the top and bottom optical path. As shown in the figure, the scattering light in the top path could go through aperture 4 and hit reflector 7, and then reflect to the corresponding quartz rod and finally reach the detector on the bottom path. This kind of interference will affect the measuring results of each optical path and lead to inaccurate measurements.

To eliminate the scattering light interference, the following model was constructed through geometric analysis to limit the distance between each optical path. As shown in Figure 5, the distance between the particle that scattered incident light and the aperture is l. The distance between the aperture and the ring-like reflector is c. The radius of the reflector is r, and the radius of the aperture is a on the narrow side and b on the wide side. The distance between adjacent optical paths is d. Under the condition that the scattered lights from different light paths did not affect each other, the scattered lights from the top light path should fall between the reflector in the middle and bottom paths. Therefore, d should satisfy the following inequalities:(12)$$\left\{\begin{array}{c} \frac{d+b}{l}\cdot \left(l+c\right)>d+r\\ \frac{d-a}{l}\cdot \left(l+c\right)<2d-r \end{array}\right.$$

By substituting the parameters of the optical structure into the inequalities, it is obtained that d≥70 mm. Considering the goal is to minimize the distance between the adjacent optical path, d=70 mm is selected. Then, the optical structure of the prototype is designed, as shown in Figure 6. The total length of the prototype is 825 mm, the width is 188 mm, and the height is 77 mm. The distance from the laser source to the measuring area is 686 mm, which can effectively separate the circuit unit and laser source from the high temperature and humidity aerosols in the measuring area. The length of the measuring area is 40 mm, and the distance between adjacent light paths is 70 mm.

### 3.2. Sampling System

In order to conduct sampling of flue smoke from a thermal power plant without distortion, the most important factor is how to eliminate the influence of steam in the flue smoke. Filters can absorb steam in the aerosol, but a filter needs to be replaced when it has absorbed to its limit. Since there is high humidity in the emitted flue smoke, the filter is not suitable for long-term work [33]. On the other hand, heating the flue smoke can convert steam into a gaseous state that makes no affection to optical measuring. To ensure that the steam converts into a gaseous state completely, it is necessary to heat all the parts from sampling to measuring to maintain the temperature above 100 °C. However, the high-temperature and high-humidity flue smoke is harmful to the circuit unit in the prototype and industrial blower, which provides pumping power for sampling. According to the optical structure designed in Section 3.1, the circuit unit is well protected by keeping a distance far from the measuring area. Similarly, to protect the industrial blower, a jetting flow sampling method based on the Venturi effect [34] has been applied to prevent the flue smoke from entering the blower. Therefore, the industrial blower is the power source of a jetting flow and has no contact with the flue smoke.

Based on the analysis of the monitoring requirements above, a sampling system has been designed. The system includes a sampling unit, a jetting flow unit, a sweeping clean air unit, a smoke parameter monitoring unit, a control unit, and the prototype. A structural diagram of the system is shown in Figure 7.

In the sampling unit, the sampling probe is inserted into the flue and fixed by a flange at a sampling port. The inlet of the probe is facing the flow direction in the flue. A heating chamber is used to heat the sampled flue smoke in order to vaporize the steam in the smoke. The measuring room is a stainless-steel box that contains the measuring area of the prototype in a vertical direction. The jetting flow unit includes a designed jetting flow pump based on the Venturi effect and an industrial blower with an air filter. The blower is controlled by the variable frequency drive (VFD) that changes the working current frequency of it. The above components are connected by sampling pipes with heat traces. The sweeping clean air unit includes a vacuum pump that blows clean air into the prototype, creating a positive pressure in the measurement room to prevent the flue smoke from entering the prototype. The smoke parameter monitoring unit includes thermocouples, temperature controllers, and a pressure transmitter for measuring the key parameters of the flue smoke. The control unit includes an industrial personal computer (IPC) to receive the data from the prototype and smoke parameter monitoring unit and control the heating chamber and VFD based on the smoke parameters measured by the monitoring unit.

The sampling process is as follows: the industrial blower blows air into the jetting flow pump and generates negative pressure in the pump to extract smoke from the flue. The smoke then goes into the heating chamber and is heated to 200°C to vaporize the steam completely. After the smoke enters the measuring room, the mass concentration and PSD are measured by the prototype. Finally, the smoke is mixed with the air in the jetting flow pump and returns to the flue through the outlet of the sampling probe. During the sampling, the temperatures of the heating chamber and measuring room are monitored. When the temperature in the measuring room drops below 120 °C, the control unit will send a command to start the heating chamber. When the temperature in the heating chamber rises above 200 °C, the control unit will stop the heating. Further, the sampling flow rate is measured by a pressure transmitter, the data are sent to the control unit, and then the VFD can change the current frequency of the blower from 30 Hz to 70 Hz in order to control the flow rate in an appropriate range.

## 4. Tests Results

The prototype and sampling system are installed on the flue of a power plant of Datang International Power Generation Co., Ltd. in Zhangjiakou, China. The flue is between the desulfurizing tower and chimney, and a platform 50 m above the ground relies on the flue. In order to protect the equipment from the cold winter in Zhangjiakou, a room was built on the platform for storing equipment and tools. Then, the equipment ran continuously for one month. Afterwards, a comparative test was held with a reference instrument to verify the multi-particle-size channel performance of the prototype.

### 4.1. Continuous Operation Test

The location of the platform and the room are shown in Figure 8. The smoke from the power plant rises in the desulfurizing tower and then enters the flue and goes down to the bottom, and then finally goes into the chimney and is emitted into the ambient environment. On the platform, there are many sampling ports along the flue for the power plant company and environmental protection department to extract and measure the emissions of the power plant, including the mass concentration of PM in the flue smoke. The monitoring data from the power plant will be uploaded to the environmental protection department simultaneously. Therefore, the results of mass concentration measured by the prototype can be compared with the data from the power plant.

Figure 9 shows the mass concentration results of PM in the flue smoke measured by the prototype compared with the monitoring data from the power plant. The monitoring data from the power plant are recorded hourly, representing the average total mass concentration during the time period. The prototype also calculates the average results in each hour for comparison. It can be seen in the figure that during 600 h of continuous measurement, the mass concentration $C_{m}$ measured by the prototype agrees with the results obtained from the power plant. The measurement error can be described as the relevant standard deviation $RSD_{C_{m}}$
(13)$$RSD_{C_{m}}=\sqrt{\frac{1}{N}{\left(\frac{M_{Pi}-M_{Ri}}{M_{Ri}}\right)}^{2}}$$
where $M_{Ri}$ is the mass concentration measured by the power plant, $M_{Pi}$ is the mass concentration measured by the prototype, and N is the sample quantity that equals the number of total hours. The calculated result of $RSD_{C_{m}}$ is only 3.617%, while the maximum ${\left[\left(M_{Pi}-M_{Ri}\right)/M_{Ri}\right]}^{2}$ is 8.274%, demonstrating a good performance of the prototype. Since the mass concentration is calculated by the hourly average, it is not surprising that the measurement error is smaller than in the laboratory test where the mass concentration is calculated by the second. The measuring error is lowered by the average of a large quantity of data.

Meanwhile, the prototype measures the mean diameter μ and the standard deviation σ of the flue smoke to obtain the distribution function $f\left(x,\mu ,\sigma \right)$. Then, the mass concentration of PM_10_, PM_2.5_, and PM_1_ can be calculated by the cumulative distribution function of volume concentration $F\left(x,\mu ,\sigma \right)$. Based on the Equation (2), $F\left(x,\mu ,\sigma \right)$ can be described as follows:(14)$$F\left(x,\mu ,\sigma \right)=\frac{1}{\sqrt{2\pi }\cdot \sigma }\int_{0}^{t}\frac{1}{x}\cdot e^{-\frac{{\left(\log x-\log\mu \right)}^{2}}{2\sigma^{2}}}\cdot \frac{\pi }{6}x^{3}dx$$
where x is the diameter of the particle and t is the upper limit of integration. Considering that the density of the flue smoke is constant, the mass concentration of PM_10_, PM_2.5_, and PM_1_ can be described as follows:(15)$$\left\{\begin{array}{c} M_{P}\left({\mathrm{PM}}_{10}\right)=\frac{1}{\sqrt{2\pi }\cdot \sigma }\int_{0}^{10000}\frac{1}{x}\cdot e^{-\frac{{\left(\log x-\log\mu \right)}^{2}}{2\sigma^{2}}}\cdot \frac{\pi }{6}x^{3}dx\\ M_{P}\left({\mathrm{PM}}_{2.5}\right)=\frac{1}{\sqrt{2\pi }\cdot \sigma }\int_{0}^{2500}\frac{1}{x}\cdot e^{-\frac{{\left(\log x-\log\mu \right)}^{2}}{2\sigma^{2}}}\cdot \frac{\pi }{6}x^{3}dx\\ M_{P}\left({\mathrm{PM}}_{1}\right)=\frac{1}{\sqrt{2\pi }\cdot \sigma }\int_{0}^{1000}\frac{1}{x}\cdot e^{-\frac{{\left(\log x-\log\mu \right)}^{2}}{2\sigma^{2}}}\cdot \frac{\pi }{6}x^{3}dx \end{array}\right.$$

The calculating result of $M_{P}\left({\mathrm{PM}}_{10}\right)$, $M_{P}\left({\mathrm{PM}}_{2.5}\right)$, and $M_{P}\left({\mathrm{PM}}_{1}\right)$ are shown in Figure 10. The mass concentrations of PM_10_, PM_2.5_, and PM_1_ change in different trends, indicating that the PSD of PM in the flue smoke varies. The change may be related to the power of the boiler, the consumption rate of oxygen during combustion, and the quality of coal. According to the general law of combustion, when the combustion is relatively slow, the combustion material will burn completely, and the particle size in the produced smoke will be smaller. On the contrary, when the combustion reaction is intense, more incomplete combustion occurs, and the particle size will be larger. In order to further analyze the relationship between PSD and combustion in flue smoke, the total mass concentration from the power plant is compared with the mean diameter μ and standard deviation σ, as shown in Figure 11 and Figure 12, respectively.

In Figure 11, the trend of mean diameter and mass concentration is basically consistent. The mean diameter increases with the rise in mass concentration, and as mass concentration declines, the mean diameter also decreases. Yet, the standard deviation is the opposite, as shown in Figure 12. When mass concentration rises, the standard deviation decreases, and when mass concentration declines, the standard deviation increases instead. When mass concentration rises, it means that the load of the boiler has increased, usually due to the peak demand for electricity. The power grid dispatches power plants to increase power generation, and the boiler needs to burn more intensively, leading to incomplete combustion. As a result, the particle size in the flue smoke increases, and the mean diameter measured by the prototype also increases accordingly. Meanwhile, the standard deviation decreases, indicating a more concentrated distribution of the particles. On the other hand, when the mass concentration decreased, it represented a decrease in boiler power, slower combustion speed, and more complete combustion, resulting in a decrease in particle size in the flue smoke and a more dispersed distribution. The necessity of measuring the PSD of PM emitted from power plants has been demonstrated. Due to the greater impact on human health from smaller particles, monitoring the PSD can comprehensively evaluate the harmful effects of PM and provide more sufficient data support for environmental supervision.

### 4.2. PM_10_, PM_2.5_, and PM_1_ Mass Concentration Test

Since the power plant only measures the total mass concentration of the flue smoke, the PSD parameters measured by the prototype are unable to be verified. Therefore, the Dekati PM_10_ impactor is applied for sampling the PM_10_, PM_2.5_, and PM_1_ on filters separately. The filters need to be thoroughly dried and weighed before sampling. During the sampling stage, the flow rate of the flue smoke is also measured in order to calculate the concentration. After sampling, the filters are dried and weighed again to obtain the mass difference. Then, the mass concentration of PM_10_, PM_2.5_, and PM_1_ is measured by calculating the flow rate and the mass difference of each filter correspondingly. A diagram of the reference instrument is shown in Figure 13.

According to the testing requirements and the practical operation status of the power plant, the test was conducted under the working load of 180 MW, 200 MW, and 250 MW. Since the time periods of each working load are different, there is only one set of data measured under 180 MW working load, three sets for 200 MW, and two sets for 250 MW. The mass concentrations $M_{0}$ of PM_10_, PM_2.5_, and PM_1_ measured by reference instrument in each sampling point are recorded, and the mean value $\bar{M_{0}}$ for each working load is calculated. The results of comparing $M_{P}$ measured by the prototype through relative error $E_{M}=\left|\bar{M_{0}}-M_{P}\right|/\bar{M_{0}}$ are shown in Table 2, and $E_{M}$ is calculated and listed.

The results show that the prototype measured the mass concentration of PM_10_, PM_2.5_, and PM_1_ correctly, according to the reference instrument’s results. The maximum relative error $E_{M}$ is 7.10%, and the average relative errors of PM_10_, PM_2.5_, and PM_1_ are 2.25%, 6.11%, and 4.28%, respectively, under different working load conditions.

The continuous operation test and PM_10_, PM_2.5_, and PM_1_ tests have proven that the prototype and the sampling system can work stably at the power plant in the long term, and the mass concentration of PM_10_, PM_2.5_, and PM_1_ can be obtained by calculating the three-wavelength scattering light signals. The measurement results show a good performance of the prototype, demonstrating the methods for comprehensive evaluation of the hazards of particulate matter emitted from a stationary source. In the end, a comparison of our prototype with the typical technologies used for continuous emission monitoring of PM is shown in Table 3 [35].

## 5. Conclusions

We proposed a method to measure the mass concentration and PSD of aerosols in thermal power plant emissions. To this end, we developed a prototype sensor based on three-wavelength technology. The prototype sensor is designed to meet the requirements of high temperature and high humidity working conditions in the thermal power plant, and its operation is tested in a practical environment in a thermal power plant in Zhangjiakou City. The test results of total mass concentration have been compared with the monitoring data from the power plant, and the relevant standard deviation is 3.617% on average during a continuous operation test for 30 days. Furthermore, the test of measuring the mass concentration of PM_10_, PM_2.5_, and PM_1_ indicates that our prototype correctly measures the multi-particle-size channel mass concentration under different working load conditions. The average relative errors of each channel are 2.25%, 6.11%, and 4.28%, respectively. In comparison with the existing apparatus employed in thermal power plants that only measures the total mass concentration using a single-wavelength laser beam, our prototype yields not only mass concentration but also the mean diameter and standard deviation of the aerosol. Therefore, the PSD of the aerosol can be determined, which is essential for the quantitative measurement of the concentration in different particle size channels. Further, the prototype sensor can be applied in other scenarios. For example, in a thermal power plant, the particle size of coal powder needs to be determined before being sent to the boiler for combustion so the prototype can measure its PSD. However, our simulation model assumes a log-normal distribution, which approximates the practical PSD. Further study on the retrieval of PSD without any preset patterns is useful for reducing the systematical error.

## Figures and Tables

**Figure 1 sensors-24-01424-f001:**
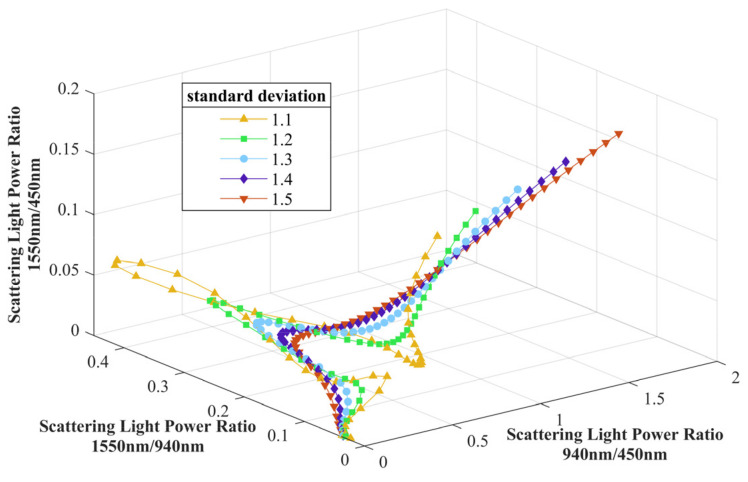
Simulation result of the scattering light power ratios.

**Figure 2 sensors-24-01424-f002:**
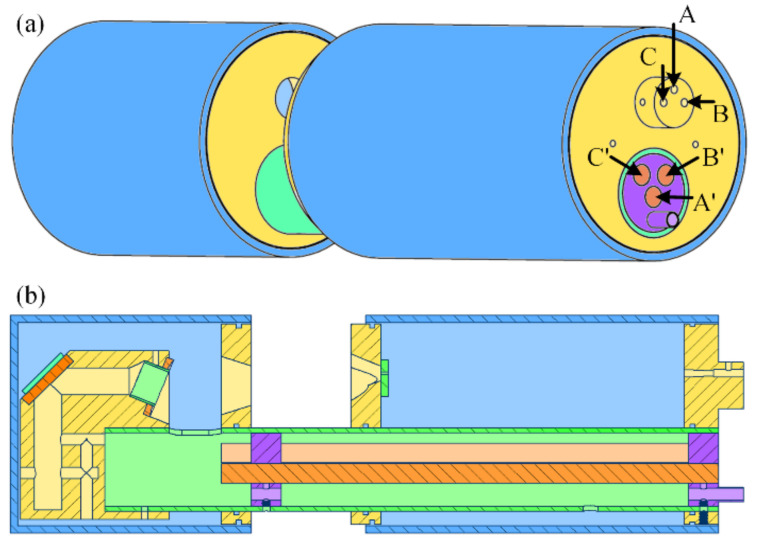
(**a**) Schematic of the appearance of the original prototype; (**b**) schematic of the internal structure of the original prototype.

**Figure 3 sensors-24-01424-f003:**
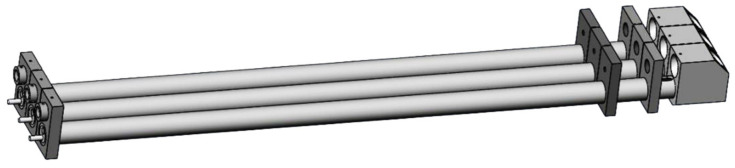
Schematic of the new optical structure without a shell.

**Figure 4 sensors-24-01424-f004:**
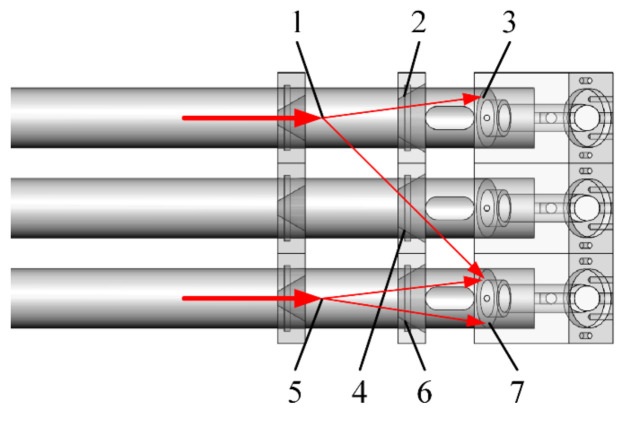
Schematic of the optical path interference.

**Figure 5 sensors-24-01424-f005:**
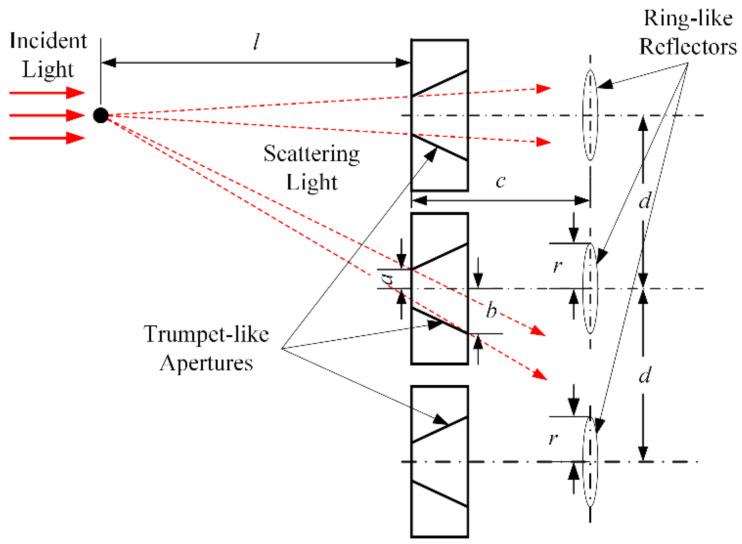
Distance between each optical path.

**Figure 6 sensors-24-01424-f006:**
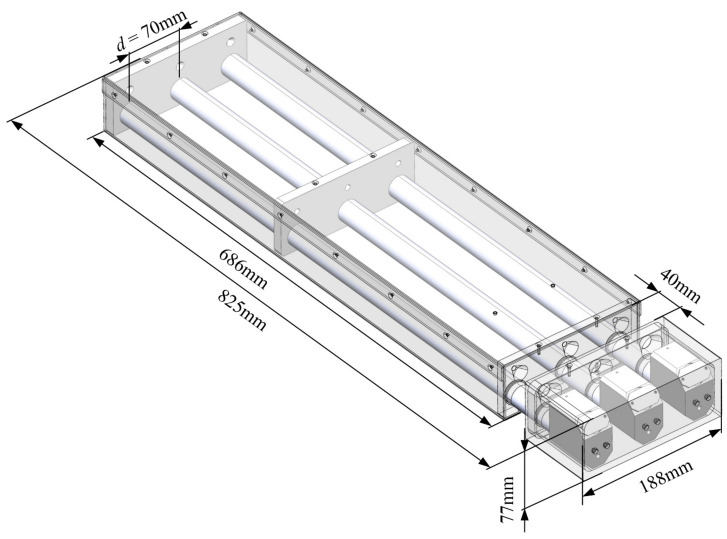
Schematic of the prototype and its parameters.

**Figure 7 sensors-24-01424-f007:**
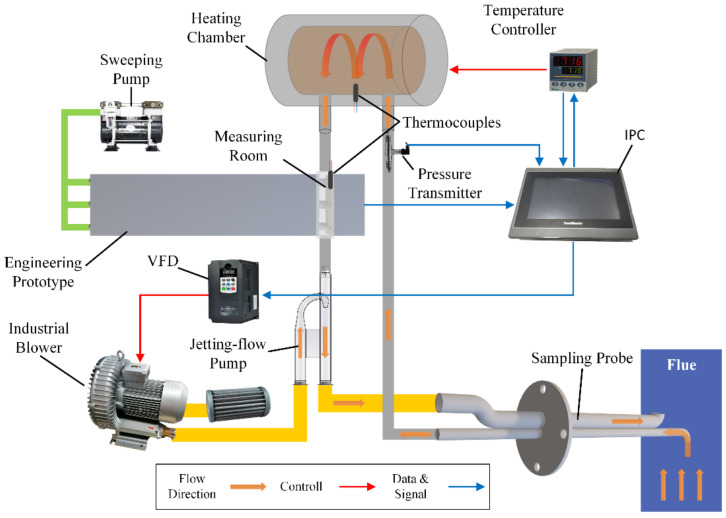
Structural diagram of the sampling system.

**Figure 8 sensors-24-01424-f008:**
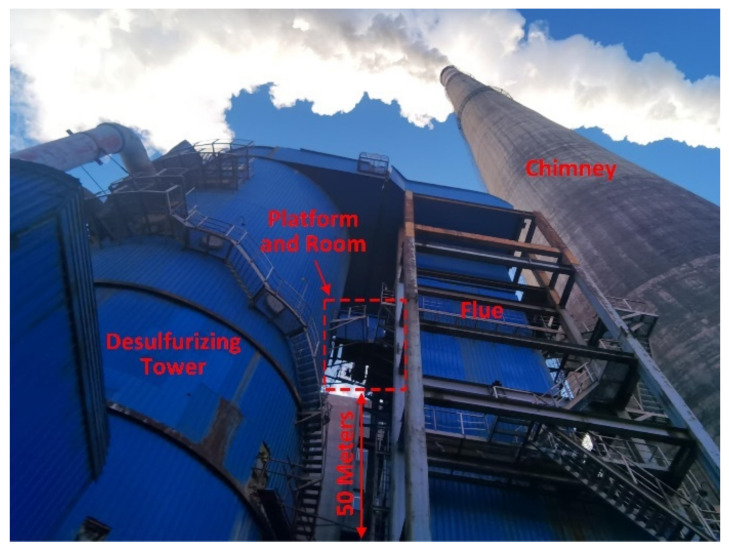
Location of the platform and room.

**Figure 9 sensors-24-01424-f009:**
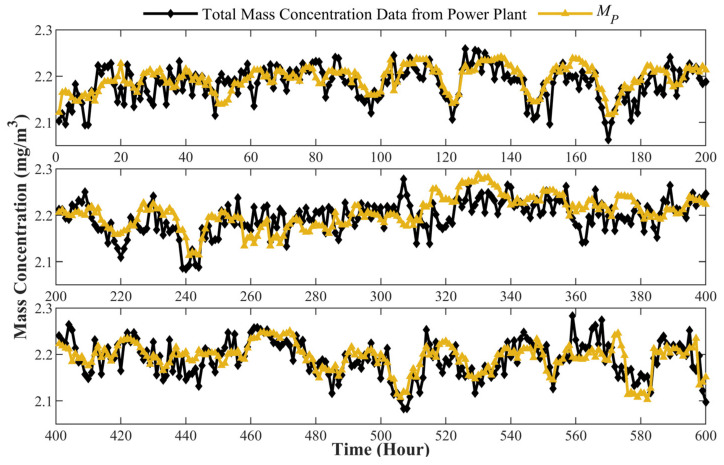
Comparison of the mass concentration measured results between data from the power plant and prototype.

**Figure 10 sensors-24-01424-f010:**
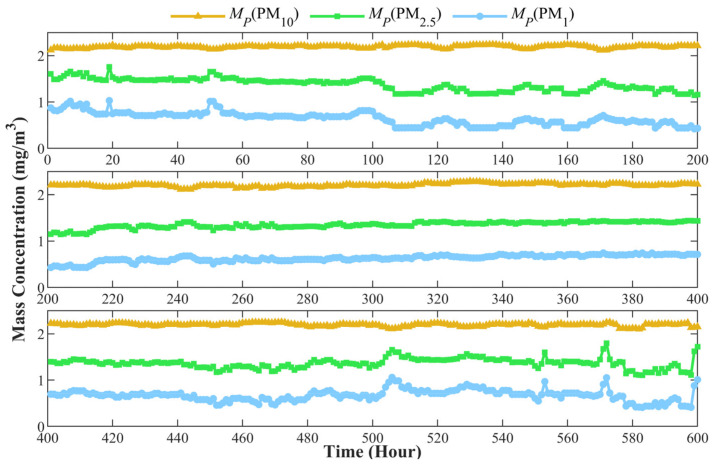
The mass concentration of PM_10_, PM_2.5_, and PM_1_ measured by prototype.

**Figure 11 sensors-24-01424-f011:**
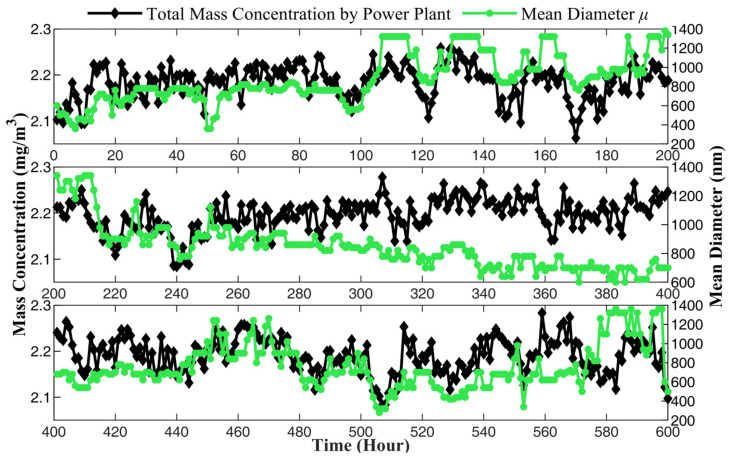
The mean diameter of the PSD measured by prototype.

**Figure 12 sensors-24-01424-f012:**
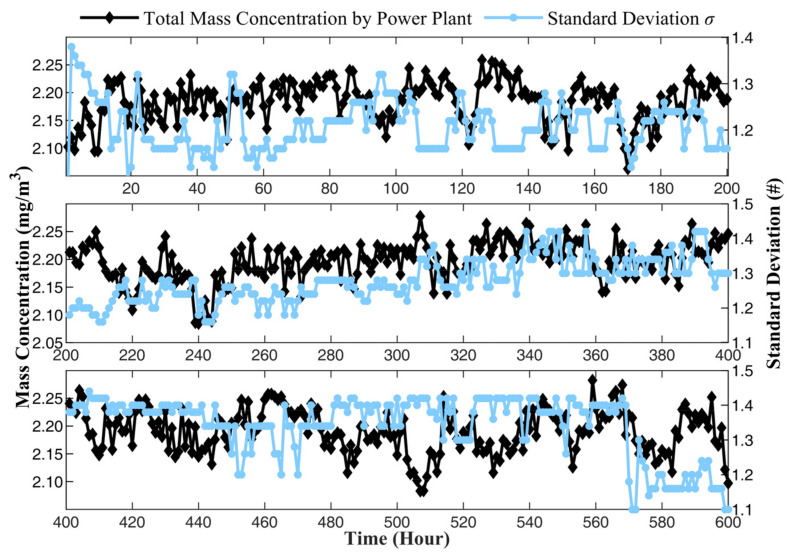
The standard deviation of the PSD measured by prototype.

**Figure 13 sensors-24-01424-f013:**
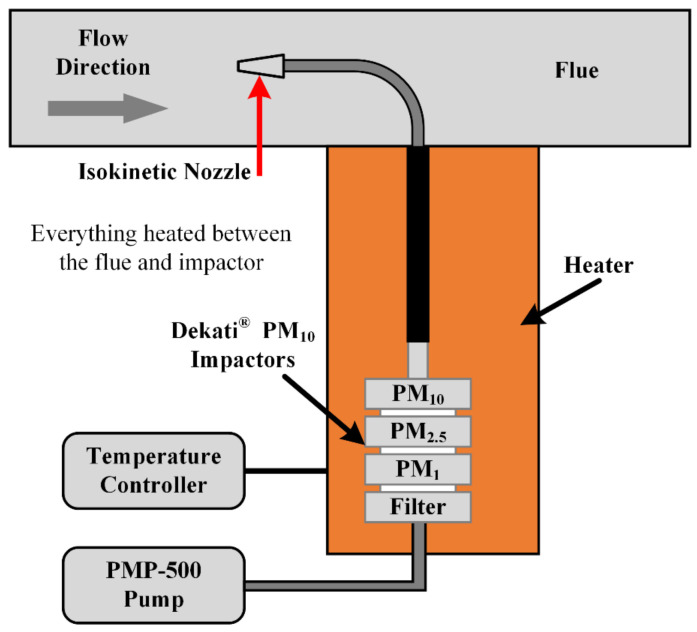
Diagram of the reference instrument, Dekati impactors.

**Table 1 sensors-24-01424-t001:** Parameter settings for the simulation.

Parameters	Range
Mean diameter μ (nm)	100:100:5000
Standard deviation σ	1.1:0.1:1.5
Wavelength of incident light (nm)	450:940:1550

**Table 2 sensors-24-01424-t002:** Comparison results of particulate matter testing in desulfurization outlet flue smoke.

Working Load	Items	$M_{0}$ (mg/m^3^)			$\bar{M_{0}}$ (mg/m^3^)	$M_{P}$ (mg/m^3^)	$E_{M}$
	Samples	P1	P2	P3	/	/	/
180 MW	PM_10_	1.70	/	/	1.70	1.68	1.20%
	PM_2.5_	1.49	/	/	1.49	1.56	4.37%
	PM_1_	0.64	/	/	0.64	0.68	6.21%
200 MW	PM_10_	1.69	1.83	1.47	1.66	1.72	3.57%
	PM_2.5_	1.54	1.67	1.32	1.51	1.61	6.84%
	PM_1_	0.77	0.95	0.59	0.77	0.80	3.51%
250 MW	PM_10_	2.38	2.25	/	2.32	2.27	1.98%
	PM_2.5_	2.06	1.83	/	1.95	2.09	7.10%
	PM_1_	1.27	0.99	/	1.13	1.09	3.11%

**Table 3 sensors-24-01424-t003:** A comparison of our prototype with the typical technologies used for continuous emission monitoring of PM.

Instrument	Principle	Distribution Measurement	Mass Concentration Range	Mass Concentration Accuracy
ESA BETA 5M ^1^	β-Ray	Not Available	0–10 mg/m^3^	0.3 μg/m^3^
LANDUN LGH-105 ^2^	TEOM	Not Available	0.1–10 mg/m^3^	0.1 μg/m^3^
SICK FWE200 ^3^	Attenuation	Not Available	0–5 mg/m^3^; 0–200 mg/m^3^	0.1 mg/m^3^; 4 mg/m^3^
Our Prototype	Light Scattering	Available	0–10 mg/m^3^; 0–250 mg/m^3^	0.1 mg/m^3^; 2.5 mg/m^3^

^1^ Environnement S.A., Poissy, France; ^2^ Anhui Landun Photoelectron Co., Ltd., Anhui, China; ^3^ Sick Engineering GmbH, Ottendorf-Okrilla, Germany.

## Data Availability

The data is unavailable due to privacy restrictions.

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
