# Peer review of "A Three-Wavelength Optical Sensor for Measuring the Multi-Particle-Size Channel Mass Concentration of Thermal Power Plant Emissions"

_sensors, 2024, doi:10.3390/s24051424_

Round 1

Reviewer 1 Report

Comments and Suggestions for Authors

(1) The original image of the prototype should be included in the manuscript.
(2) The authors are encouraged to add real-time measurement images of the dust concentrations.
(3) The authors are encouraged to decide the danger level of the measured date based on WHO standard. The authors can follow the style of the following paper if they want.

I. -B. Sohn, H. -K. Choi, Y. -J. Jung, C. -J. Lee, M. -K. Oh and M. S. Ahsan, "Measurement of Fine/Ultrafine Dust Using Lenticular Fiber-Based Particulate Measurement Devices," in IEEE Sensors Journal, vol. 23, no. 8, pp. 8400-8409, 15 April15, 2023, doi: 10.1109/JSEN.2023.3251367.

(4) The authors are encouraged to compare their results with the measurements obtained by a commercial device. 

Reviewer 2 Report

Comments and Suggestions for Authors

Authors conducted a good work on the detection of particulate matter in thermal power plant. The optical sensor with three wavelength has the great potential for application. It is necessary to provide more detailed explanations about the following content.

1. What are the external conditions for the stable operation of this optical sensor? What are the effects of different weather conditions on the test results?

2. Please provide a table to compare the advantages and disadvantages of existing measurement methods with this work. For example, sensitivity, error, anti-interference, portability, cost, etc.

3. For this type of optical sensor, please list other applications and provide a brief outlook.

Comments on the Quality of English Language

Minor editing of English language is required.

Reviewer 3 Report

Comments and Suggestions for Authors

In this manuscript, the authors presented a novel detection method for particle based on three-wavelength technique to measure the mass concentrations of PM10, PM2.5 and PM1 simultaneously. First, the authors designed an optical sensor to align three optical paths with low interference between each optical path. A mathematical model was then developed to analyze the signals and to evaluate the particle concentrations. The optical sensor was applied and tested in a practical thermal power plant, and the results showed that the sensor can correctly measure the mass concentration in multi-particle-size channels with acceptable errors. Overall, the article is well written with good figure organization. It is thus recommended for publication. However, some minor issues need to be addressed before publication and are listed below:

1.      In the introduction, the authors only mentioned the particle size issues. But the difficulties of particle size measurements are not mentioned. Why is it necessary to measure with multiple wavelengths? Moreover, the authors should provide some background why need to measure particle sizes at 1, 2.5, and 10 micrometers.

2.      The incident light wavelengths are not well described. Why using these three wavelengths in Table 1? Is there a general guideline between the incident wavelength with the particle sizes?

3.      In the test results, the authors provided the data comparison between standard and the current design. The authors also claimed good accuracy in table 2. But as the authors mentioned in the introduction, the weather can also cast some influences on the measurements. Is it possible to provide the weather data, such as temperature or humidity, along with the concentration measurements?

Reviewer 4 Report

Comments and Suggestions for Authors

In the article, the authors consider the experimental implementation of their own method for measuring the concentration of particles with a certain size distribution. However, in the form that the authors presented, it is difficult to understand the purpose of the work. 

1. In the article on the verification of the method, the authors refer the reader to their earlier work with a link [29] (this is an error, probably it is necessary [30]). The authors in this paper do not have the opportunity to compare the results related to the particle size distribution!

2.  The authors did not confirm in any way the importance of information on the distribution of particle sizes in industrial emissions on human health. The review needs to be improved to confirm the importance of this information!

3. In my opinion, Fig. 8, Fig.10, Fig.11 are completely unsuitable for illustrating the results. The authors should divide them into several shorter sections so that not only reviewers but also readers have the opportunity to evaluate the results.

Round 2

Reviewer 1 Report

Comments and Suggestions for Authors

The revised manuscript can be accepted.

Reviewer 2 Report

Comments and Suggestions for Authors

Revisions are OK.

Comments on the Quality of English Language

Minor editing of English language is required.

Reviewer 4 Report

Comments and Suggestions for Authors

The authors took my comments into account. The new version of the article looks much more logical.